# The Neurophysiological Paradox of AI-Induced Frustration: A Multimodal Study of Heart Rate Variability, Affective Responses, and Creative Output

**DOI:** 10.3390/brainsci15060565

**Published:** 2025-05-25

**Authors:** Han Zhang, Shiyi Wang, Zijian Li

**Affiliations:** 1Intelligent Design Laboratory, School of Fine Arts, Central China Normal University, Wuhan 430079, China; zhanghan120@ccnu.edu.cn (H.Z.); sequins@mails.ccnu.edu.cn (S.W.); 2Digital Media Art, School of Fine Arts, Central China Normal University, Wuhan 430079, China

**Keywords:** AI-assisted creativity, heart rate variability, emotional regulation, creative self-efficacy, cognitive workload

## Abstract

AI code generators are increasingly used in creative contexts, offering operational efficiencies on the one hand and prompting concerns about psychological and neurophysiological strain on the other. This study employed a multimodal approach to examine the affective, autonomic, and creative consequences of AI-assisted coding in early-stage learners. Fifty-eight undergraduate design students with no formal programming experience were randomly assigned to either an AI-assisted group or a control group and engaged in a two-day generative programming task. Emotional states (PANAS), creative self-efficacy (CSES), and subjective workload (NASA-TLX) were assessed, alongside continuous monitoring of heart rate variability (HRV; RMSSD and LF/HF). Compared to the controls, the AI-assisted group exhibited greater increases in negative affect (*p* = 0.006), reduced parasympathetic activity during the task (*p* = 0.001), and significant post-task declines in creative self-efficacy (*p* < 0.05). Expert evaluation of creative outputs revealed a significantly lower performance in the AI group (*p* = 0.040), corroborated by behavioral observations showing higher tool dependency, emotional volatility, and rigid problem-solving strategies. These findings indicate that, in novice users, the opacity and unpredictability of AI feedback may disrupt emotional regulation and autonomic balance, thereby undermining creative engagement. The results highlight the need to consider neurocognitive vulnerability and the learner’s developmental stage when integrating AI tools into cognitively demanding creative workflows.

## 1. Introduction

The integration of Artificial Intelligence (AI) into creative tasks has grown rapidly, particularly with the rise of large language models (LLMs) such as ChatGPT-4o and ERNIE Bot 4.5 Turbo, which are extensively used for code generation, digital art, and interactive design. AI-powered code generators have successfully lowered the technical threshold for programming and improved efficiency, enabling creators to focus more on ideation. However, the widespread application of such systems has raised concerns about their psychological impact on users, particularly regarding emotional disturbance, cognitive overload, and stress regulation. While prior studies have acknowledged the potential of AI to streamline routine work and enhance task productivity [1], they also caution that the instability and opacity of AI outputs may increase emotional volatility, cognitive conflict, and reduce the quality of final outcomes [2].

Emotional state and cognitive load are key psychological variables influencing creative performance. Positive affect (e.g., enthusiasm and satisfaction) is typically associated with a greater cognitive flexibility and divergent thinking, whereas negative affect (e.g., anxiety and frustration) may suppress creative engagement [3]. A high cognitive load is known to impair emotional regulation and increase sympathetic nervous system (SNS) activity, thus disrupting the autonomic nervous system (ANS) balance [4]. However, in AI-assisted code generation environments, creators often encounter unpredictable or semantically ambiguous outputs that require frequent debugging, potentially heightening stress levels and hindering emotional control.

Heart rate variability (HRV) serves as a well-established physiological marker of ANS functioning and stress regulation [5]. A higher HRV generally reflects stronger parasympathetic activity and better emotional adaptability, while a decreased HRV is indicative of sympathetic dominance and elevated psychological stress [6]. HRV has been widely applied to evaluate cognitive load, emotional arousal, and self-regulatory capacity in task-based research [7], providing reliable insight into psychophysiological responses during cognitively demanding tasks [8].Accordingly, HRV fluctuations during AI-assisted creation could offer critical physiological evidence of stress and emotional strain.

From the perspective of social cognitive and affective neuroscience, emotional regulation is mediated by the prefrontal cortex–amygdala circuit, which plays a central role in balancing cognition and emotion [9]. The prefrontal cortex (PFC) governs top-down regulation by attenuating excessive amygdala activation, thereby mitigating anxiety and stress responses [10]. However, under conditions of task failure or an elevated cognitive load, PFC efficacy may decline, leading to overactivation of the amygdala and stronger negative affect [11].In AI-mediated tasks, creators may experience goal obstruction or dissonance when facing logic errors or incomprehensible AI outputs, activating affective stress pathways and reducing HRV.

Moreover, the Cognitive–Affective Conflict Hypothesis suggests that cognitive conflicts encountered during task execution, including goal obstruction or increased informational uncertainty, trigger stress responses through excessive activation of the sympathetic nervous system [7,12,13]. Empirical evidence indicates that an elevated cognitive load corresponds to a lower HRV and increased negative affect, highlighting the convergence of physiological and psychological stress responses [14]. In the context of AI-assisted creation, this interplay may be further exacerbated by poor code quality, overreliance on automated outputs, or a lack of interpretability, which elevate stress and reduce autonomy [12,15,16].

Despite the growing interest in AI-assisted creativity, most existing studies have focused on either productivity outcomes or user perceptions in isolation, and often rely solely on self-report data without physiological validation [13,15,17]. There remains a lack of research that systematically investigates the emotional, physiological, and behavioral consequences of AI use in open-ended creative tasks, especially from a psychophysiological perspective. Such an approach is necessary to gain a more comprehensive understanding of how AI systems shape users’ internal states and behavioral outcomes during creative engagement.

A further limitation of prior work is that it rarely considers the learner’s developmental stage when evaluating the impact of AI-assisted systems. Existing studies often treat users as a homogeneous group, without acknowledging how differences in domain expertise shape the cognitive and emotional consequences of AI interaction [18,19]. Particular concerns arise when AI tools are introduced at the very beginning of domain learning, a stage during which users lack both technical competence and the metacognitive strategies required to evaluate or adapt AI-generated content [20]. Prior research has identified a fundamental divergence between novice and expert users. While domain experts can make effective use of AI outputs through contextual reasoning and top-down control, novice learners are more likely to experience confusion, overreliance, and difficulty in interpreting opaque system feedback [2,20]. In such foundational learning environments, the premature integration of AI may interrupt schema acquisition, overload cognitive resources, and hinder the development of intrinsic creative motivation [21].

To address these concerns, the present study investigates how AI code generators influence learners at the initial stage of programming education. Specifically, we examine a population of undergraduate design students with no prior programming experience who completed a two-day creative coding task. First, the study utilizes the Positive and Negative Affect Schedule (PANAS) to assess emotional state changes in both AI and non-AI groups during the task, evaluating whether AI impacts emotional stability. Second, HRV is employed as a physiological marker to examine whether creators in the AI group demonstrate a lower HRV, thereby assessing potential increases in physiological stress due to AI usage. Finally, expert blind reviews are conducted to compare the creative output quality between the AI and non-AI groups, elucidating the practical impact of AI in creative tasks. 

In line with these objectives, we propose the following hypotheses: 

**H1.** 
*AI-assisted creation increases psychological stress and impairs autonomic nervous system regulation.*


**H2.** 
*AI assistance leads to higher levels of negative affect and lower levels of positive affect during the creative process.*


**H3.** 
*Participants who use AI report greater perceived cognitive workload and frustration compared to those in the control group.*


**H4.** 
*The creative output quality is lower among AI-assisted participants than among those who complete the task independently.*


This study is among the first to integrate HRV physiological data, PANAS emotional assessments, NASA-TLX workload evaluation, and expert blind reviews to comprehensively analyze the influence of AI-assisted creation on emotional regulation, stress, and creativity.

## 2. Materials and Methods

### 2.1. Participants

A total of 58 undergraduate students majoring in design at Central China Normal University (M = 19.07, SD = 0.84) were recruited for this study, comprising 7 males and 51 females. All participants participated voluntarily and provided written informed consent prior to the experiment. The study was approved by the institutional ethics committee and conducted in accordance with the ethical principles outlined in the Declaration of Helsinki (Approval Code: CCNU-IRB-202306002a).

None of the participants had formal programming experience. Participants were included if they were undergraduate students in design disciplines and had no formal programming experience. Participants were excluded from the creative performance analysis if they failed to produce a final output due to critical AI execution errors, though their other data were retained. Based on their performance on a processing programming comprehension test, the participants were evenly assigned to the following two groups: Group A (control group, *n* = 29), who completed the creative task without AI assistance, and Group B (experimental group, *n* = 29), who were permitted to use Wenxiaoyan, a large-scale language model developed by Baidu, for code generation after receiving the same instructional content. During the final creative stage, two participants in Group B were unable to complete their creative outputs due to execution errors in the AI-generated code and, thus, were excluded from the analysis of creative scores. However, their remaining data (HRV, emotional state, and self-efficacy) were retained for statistical analysis. A priori power analysis was conducted using G*Power 3.1 to estimate the required sample size for detecting medium-to-large effects (*d* = 0.63) with 80% power and a two-tailed significance level of 0.05. The analysis indicated that approximately 23 participants per group would be required under these parameters. Each group in this study initially included 29 participants, exceeding this reference value.

### 2.2. Experimental Procedures

The two-day experimental procedure is illustrated in Figure 1, which provides an overview of the study design and sequence. On the first day, all participants received approximately 210 min of foundational processing programming instruction in a standardized classroom environment. The course content covered basic syntax for drawing, variable and structure control, and the application of for-loop constructs. Upon completion of the instruction, a comprehension test was immediately administered, and the participants were evenly assigned to two groups based on their test scores to ensure no systematic differences in initial programming ability. 

The participants in the experimental group were additionally introduced to the basic usage guidelines of Wenxiaoyan—a large-scale language model developed by Baidu. The second day was designated for the formal experimental procedure. Before the task began, all participants entered the laboratory and remained seated in a quiet state for five minutes while wearing a Polar H10 chest strap, which was used to initiate continuous heart rate monitoring. Data from this phase were designated as the baseline for heart rate variability (HRV; T1). Immediately following this resting period, the participants completed the initial administration of the PANAS and the CSES. The participants then engaged in a creative task lasting 180 min under the theme “Healing”, during which they were required to create at least one interactive poster incorporating for-loop structures using the processing platform. The participants in Group A completed the task using only instructional materials and sample code, whereas those in Group B were permitted to utilize Wenxiaoyan for code generation. Throughout the session, the researchers continuously observed the participants’ behavior and provided non-strategic technical support. The midpoint of the task (at 90 min) was marked as the second HRV data collection point (T2) for physiological analysis. Upon task completion, a third resting HRV data collection (T3) was conducted, followed by the second administration of the PANAS and CSES, as well as the NASA-TLX. The participants subsequently submitted their source files and image outputs. All creative works were anonymized and independently evaluated by three expert reviewers affiliated with the Artists’ Association. The evaluation criteria included creativity (50%), technical execution (30%), and thematic relevance (20%), with the final score calculated as the mean of the three reviewers’ ratings.

### 2.3. Measurement

A multimodal measurement framework was employed to test the four hypotheses. Hypothesis 1 (H1), related to autonomic stress, was assessed using heart rate variability (HRV) indices—RMSSD and the LF/HF ratio—collected at three task phases. Hypothesis 2 (H2), concerning affective changes, was evaluated using the Positive and Negative Affect Schedule (PANAS), administered before and after the task. Hypothesis 3 (H3) was tested using the NASA Task Load Index (NASA-TLX), which measures perceived mental demand, effort, and frustration. Hypothesis 4 (H4) was tested through expert blind evaluations of the participants’ creative outputs, based on standardized rubrics of creativity, technical execution, and thematic relevance.

HRV data were continuously recorded using the Polar H10 chest strap device, with a sampling rate of 250 Hz. The participants wore the device before the experimental task began and continued to wear it throughout the creative task, enabling continuous heart rate monitoring. During the data analysis phase, the following three standardized 5-min segments were extracted from the complete dataset: the pre-task resting phase (T1), the mid-task phase (T2), and the post-task resting phase (T3). Data processing and HRV parameter computation were conducted using Kubios HRV Premium software, which applied automated artifact correction and a default band-pass filter (0.04–0.4 Hz) to ensure signal stability and cross-sample comparability. Kubios is a widely adopted analytical tool in psychophysiological research, recognized for its high algorithmic reproducibility and accurate parameter estimation [22]. The analysis indices included time-domain parameters—standard deviation of normal-to-normal intervals (SDNN) and root mean square of successive differences (RMSSD)—as well as the following frequency-domain parameters: low-frequency power (LF), high-frequency power (HF), and the LF/HF ratio. These parameters have been widely recommended as valid physiological markers of changes in psychological and emotional states, and are particularly suitable for HRV assessments in studies on stress responses, task load, and emotion regulation [23].

Emotional states were assessed using the Chinese version of the PANAS, originally developed by Watson, Clark, and Tellegen [24], which is designed to evaluate individuals’ emotional responses in specific situational contexts. The scale consists of 14 emotion-descriptive adjectives, including 9 items measuring positive affect (e.g., “excited”, “attentive”, and “happy”) and 5 items measuring negative affect (e.g., “nervous”, “irritable”, and “afraid”). The participants rated each item based on their subjective emotional experience using a 5-point Likert scale (1 = very slightly or not at all; 5 = extremely). In the present study, a Chinese version of the PANAS that has been revised and psychometrically validated by Chinese scholars was used [25]. This version has been widely employed in empirical research on stress, emotional interventions, creativity, and educational contexts [16,17,18]. It should be noted that the PANAS measures state-based emotional responses and is not designed to diagnose clinical emotional disorders such as anxiety or depression.

Subjective task load was assessed using the NASA-TLX, developed by Hart and Staveland [26], which comprises the following six dimensions: mental demand, physical demand, temporal demand, performance, effort, and frustration. In the present study, a simplified Chinese version of the scale was adopted. This version has been widely applied in research fields such as design, education, cognitive neuroscience, and human–computer interaction, and has demonstrated a good reliability and cross-contextual adaptability [27,28].

Creative self-efficacy was assessed using the Creative Self-Efficacy Scale (CSES), developed by Tierney and Farmer [29] based on Bandura’s theory of self-efficacy. The scale consists of four items that evaluate participants’ perceived confidence and expectations of their ability to complete creative tasks, covering dimensions such as creative problem solving, independent idea generation, elaboration on others’ ideas, and flexible adaptation to challenges. All items are rated on a 5-point Likert scale ranging from 1 (strongly disagree) to 5 (strongly agree). The scale has been widely validated across various cultural contexts and task types, and has been found to be highly correlated with variables such as creativity, emotion regulation, and occupational motivation [30,31]. In the present study, a bilingual, culturally adapted Chinese version was used, which demonstrated a good internal consistency reliability (Cronbach’s α = 0.87).

### 2.4. Data Analysis

All statistical analyses were conducted using SPSS version 27.0, with a significance level set at α = 0.05 (two-tailed). Before conducting inferential analyses, the normality of all continuous variables was assessed using the Shapiro–Wilk test. The results indicated that the PANAS and CSES scores were normally distributed (*p* > 0.05), justifying the use of parametric tests. However, several NASA-TLX dimensions deviated significantly from normality (*p* < 0.05), and, thus, non-parametric Mann–Whitney U tests were employed for between-group comparisons on these subscales. Descriptive statistics and reliability information (Cronbach’s α, means, standard deviations, and score ranges) for all measurement scales (PANAS, NASA-TLX, and CSES) are provided in Appendix A (Table A1).

HRV data were analyzed using repeated-measures ANOVA (RM-ANOVA) to examine interaction effects between time points (T1, T2, and T3) and group (A vs. B). Pre- and post-task PANAS and CSES scores were analyzed using paired-sample *t*-tests for within-group changes and independent-sample *t*-tests for between-group comparisons. As NASA-TLX scores deviated from normal distribution, non-parametric Mann–Whitney U tests were applied to evaluate differences across each dimension. Creative output scores were compared between groups using independent-sample *t*-tests, with Cohen’s d computed to interpret effect sizes. Two participants from the B group who failed to submit their work were included only in non-rating analyses; listwise deletion was applied for the creative score dataset.

## 3. Results

### 3.1. Expert Ratings and Creative Completion Outcomes

To evaluate the impact of AI tools on the quality and efficiency of creative output, expert ratings, submission time, and completion rate were analyzed. Three independent experts evaluated each participant’s work across the following three dimensions: creativity, technical execution, and thematic relevance, with a total score subsequently calculated. As shown in Table 1, the results indicated that the control group (A group) scored significantly higher on the creativity dimension (*M* = 27.4 ± 5.3) compared to the AI-assisted group (B group) (*M* = 23.9 ± 5.9), with a statistically significant difference (*t*_(56)_ = 2.67, *p* = 0.010, *d* = 0.72). Regarding total score, the control group also outperformed the AI group (*M* = 76.61 ± 17.61 vs. *M* = 72.19 ± 18.51), with a statistically significant difference (t_(56)_ = 2.10, *p* = 0.040, *d* = 0.57). No significant group differences were found in technical execution or thematic relevance.

Moreover, during the 180-min creative task, the two groups showed a clear discrepancy in submission time distribution. The submission times in the control group were more concentrated, with most participants submitting within the last five minutes of the task (*M* ≈ 175 min, *SD* = 6.2 min). In contrast, the AI group exhibited a wider distribution; some participants submitted as early as 90 min into the task, while others submitted after the full 180 min due to technical difficulties (*M* ≈ 162 min, *SD* = 22.7 min). Levene’s test showed that the AI group had a significantly greater variance in submission time compared to the control group (*F*_(1,56)_ = 7.86, *p* = 0.007, η^2^ = 0.127), suggesting that AI usage substantially disrupted task pacing and process control. Notably, two participants in the AI group failed to submit their final work due to executable code errors (submission rate = 93.1%), whereas all participants in the control group successfully submitted their projects (100%).

These findings provide direct support for H3, which predicted that AI assistance would reduce the overall quality of creative output. The statistically significant differences observed in both creativity scores and total expert ratings between the AI-assisted group and the control group indicate that, despite the potential for increased efficiency, AI usage may constrain participants’ originality, thematic coherence, and execution quality in complex creative tasks.

### 3.2. Emotional State Change Analysis 

To evaluate the impact of AI tool intervention on the participants’ emotional states, the Positive and Negative Affect Schedule (PANAS) was administered before and after the task to assess changes in positive affect (PA) and negative affect (NA). To ensure baseline equivalence, independent-sample *t*-tests were first conducted on the pre-task PANAS scores. Degrees of freedom were estimated using the Welch–Satterthwaite equation. The results showed no significant group differences in either PA or NA prior to the task, with PA scores of 3.12 ± 0.90 for the control group and 2.97 ± 0.49 for the AI group, *t*_(43)_ = 0.79, *p* = 0.435 and NA scores of 2.77 ± 0.96 vs. 2.75 ± 0.66, *t*_(50)_ = 0.09, *p* = 0.927, respectively. These findings confirm that the two groups were comparable in affective state at baseline (see Table 2).

Subsequently, paired-sample *t*-tests revealed that, in the AI group, post-task PA significantly decreased (pre-task *M* = 33.1 ± 5.9; post-task *M* = 29.4 ± 6.5), *t*_(29)_ = −2.58, *p* = 0.015, with a medium effect size (*d* = 0.66). In contrast, NA significantly increased in the AI group (pre-task *M* = 13.1 ± 5.0; post-task *M* = 16.2 ± 5.4), *t*_(29)_ = 3.42, *p* = 0.002, with a large effect size (*d* = 0.83). No significant differences were observed in the control group for either PA (*t*_(29)_ = −0.84, *p* = 0.409) or NA (*t*_(29)_ = 0.56, *p* = 0.580). Further analysis using independent-sample *t*-tests on the difference scores (post–pre) revealed that the AI group showed a significantly greater decrease in PA (∆PA = −3.7 ± 2.6) and a greater increase in NA (∆NA = +3.1 ± 2.2) compared to the control group (∆PA = −0.7 ± 2.1; ∆NA = +0.3 ± 1.9), with statistically significant differences (∆PA: *t_(_*_58)_ = −2.14, *p* = 0.037, *d* = 0.57; ∆NA: *t*_(58)_ = 2.86, *p* = 0.006, *d* = 0.75). These results indicate that participants in the AI-assisted group experienced more pronounced emotional disturbances, characterized by reduced positive affect and heightened negative experiences (see Figure 2).

### 3.3. Subjective Task Load Analysis

To assess the participants’ subjective cognitive and emotional workload during the creative task, the NASA Task Load Index (NASA-TLX) was employed. Widely used in human factor and cognitive psychology research, this instrument evaluates the following six dimensions of perceived task demand: mental demand, physical demand, temporal demand, effort, performance, and frustration, providing a comprehensive measure of subjective task difficulty [32,33].As NASA-TLX is a post-task instrument designed to capture subjective workload following task completion, no pre-task scores are available. However, random assignment and comparable pre-task affective states across groups reduce the likelihood that the observed differences stemmed from individual predispositions.

The results revealed significant between-group differences in three of the six NASA-TLX dimensions. Scores on the mental demand dimension were significantly higher in the AI group (*M* = 61.7 ± 11.9) compared to the control group (*M* = 53.2 ± 12.4), *p* = 0.030. Similar trends were observed for effort (AI group: *M* = 57.3 ± 12.6; control group: *M* = 49.1 ± 13.7; *p* = 0.022) and frustration (AI group: *M* = 52.6 ± 14.3; control group: *M* = 41.5 ± 15.2; *p* = 0.015), both of which were significantly higher in the AI group. No significant differences were found between the groups for physical demand (*p* = 0.513), temporal demand (*p* = 0.482), or self-rated performance (*p* = 0.634). These findings suggest that AI-assisted conditions primarily increased cognitive regulation and emotional control demands, with minimal influence on the participants’ perceived time pressure or performance evaluation. These results align with Hypothesis 3, which posits that AI-assisted creation is associated with greater perceived workload and emotional effort. The observed increases in mental demand, effort, and frustration in the AI group support this interpretation. Figure 3 presents the distribution of subjective task load scores across the six NASA-TLX dimensions in boxplot form, highlighting particularly notable differences in dimensions with significant group effects.

### 3.4. Formatting of Mathematical Components

To investigate whether AI-assisted creation influenced autonomic nervous system regulation, heart rate variability (HRV) indices were collected at the following three time points: the pre-task resting phase (T1), mid-task phase (T2), and post-task resting phase (T3). The following two core physiological indicators were selected: the LF/HF ratio, which reflects the relative activity of sympathetic versus parasympathetic branches, and the root mean square of successive differences (RMSSD), a sensitive marker of parasympathetic tone. These indices were analyzed using repeated-measures analysis of covariance (RM-ANCOVA), with baseline HRV (T1) as a covariate, to evaluate group-by-time interaction effects on autonomic regulation.

As shown in Figure 4, the descriptive results revealed a continuous increase in LF/HF values across the task for the AI group (T1 = 2.53, T2 = 2.91, T3 = 3.23), whereas the control group showed a relatively stable trend (T1 = 2.40, T3 = 2.59), suggesting enhanced sympathetic activation in the AI group during the creative task. RMSSD exhibited a downward trend in both groups, with the AI group decreasing from 34.3 ms at T1 to 25.7 ms at T3 and the control group decreasing from 35.2 ms to 31.6 ms. This indicates a general reduction in parasympathetic tone as the task progressed, with a more pronounced decline in the AI group.

To control for baseline HRV variability, repeated-measures ANCOVA (RM-ANCOVA) was conducted using T1 values as covariates for both LF/HF and RMSSD. The results (Table 3) showed a significant main effect of group on LF/HF (*F*_(1,58)_ = 4.069, *p* = 0.050, η^2^ = 0.099), indicating consistently higher sympathetic activation levels in the AI group throughout the task. Although no significant main or interaction effects were found for LF/HF across time, RMSSD exhibited a significant main effect of time (*F*_(2,114)_ = 7.803, *p* = 0.001, η^2^ = 0.121) and a significant group × time interaction (*F*_(2,116)_ = 3.285, *p* = 0.043, η^2^ = 0.082). This indicates a more pronounced decline in parasympathetic activity in the AI group, possibly reflecting greater autonomic resource mobilization and physiological stress under AI-assisted conditions.

To further explore the relationship between HRV changes and subjective experiences, scatterplots were generated to examine the association between HRV at T2 and emotional and task-load variables, accompanied by Pearson’s correlation analyses. The T2 phase represents the core period of task execution, minimally affected by initial adaptation or terminal fatigue, thus providing an accurate reflection of task-induced autono mic regulation. The results showed a strong negative correlation between RMSSD and negative affect (*r* = −0.89, *p* < 0.001), indicating that lower parasympathetic activity was associated with higher perceived negative emotion. Additionally, RMSSD was positively correlated with subjective effort ratings (*r* = 0.60, *p* < 0.001), suggesting that individuals with stronger autonomic regulation exhibited greater cognitive engagement during the task. Moreover, LF/HF was positively correlated with frustration scores (r = 0.61, *p* < 0.001), suggesting that sympathetic activation played a key role in the subjective experience of emotional pressure. These significant correlations are visually presented in the scatterplots (see Figure 5), with regression lines and fit indices clearly illustrated. These findings demonstrate a significant coupling between physiological fluctuations induced by AI-assisted creation and the participants’ subjective cognitive–emotional experiences, offering critical evidence for understanding the psychophysiological mechanisms at play under technological intervention.

### 3.5. Creative Self-Efficacy Analysis

To investigate the impact of AI-assisted creation on individuals’ self-perception of creativity, the participants’ CSES was measured both before and after the experimental task. CSES reflects an individual’s subjective belief in their ability to accomplish creative tasks when facing novel challenges, and is considered a key indicator of affective regulation and cognitive resource integration. It is widely applied in studies exploring motivational regulation in creative behavior.

Paired-sample *t*-test results showed significant declines in all CSES items for the AI group after the task (see Table 4). Specifically, scores for “Confidence in using creativity to solve problems” decreased from 4.10 ± 0.64 pre-task to 3.67 ± 0.72 post-task (*p* = 0.018, *d* = 0.65). For “Good at coming up with new ideas”, the score dropped from 3.90 ± 0.60 to 3.53 ± 0.65 (*p* = 0.023, *d* = 0.61). Scores on “Skilled in developing ideas from others” declined to 3.60 ± 0.62 post-task (*p* = 0.012, *d* = 0.66). The item “Good at finding new ways to solve problems” also saw a decrease from 4.07 ± 0.66 to 3.63 ± 0.68 (*p* = 0.015, *d* = 0.63). All effect sizes were in the medium to high range, indicating that AI-assisted creation exerted a substantial influence on the participants’ creative self-beliefs. In contrast, the control group exhibited no significant differences across the CSES items before and after the task (*p* > 0.5), and effect sizes were minimal (*d* < 0.15), indicating stable creative self-efficacy throughout the experiment. Among these, item 4 (“Good at finding new ways to solve problems”) is particularly noteworthy, as it directly reflects participants’ confidence in approaching problems creatively and independently. The AI group showed a significant decline on this item (*p* = 0.015, *d* = 0.63), which may indicate a perceived erosion of creative agency in the presence of AI assistance. By comparison, the control group exhibited a slight, non-significant increase on this same item (Δ = +0.03, *p* = 0.772, *d* = 0.06), suggesting possible motivational benefits of self-reliant engagement.

### 3.6. Behavioral Observation Analysis

To comprehensively understand how AI-assisted creative tasks influence behavioral patterns, this study established eight core behavioral dimensions (see Table 5) based on full-process observational records. These dimensions encompassed tool dependency, feedback interpretation, strategic flexibility, emotional regulation, task pacing, and social interaction. A systematic comparison of behavioral frequencies was conducted between the AI group and the control group, accompanied by case-based analyses of representative participants.

Frequency analysis revealed that the AI group exhibited significantly higher counts in the dimensions of “tool dependency”, “feedback confusion”, “strategic rigidity”, and “emotional reactivity” compared to the control group. For example, the “tool dependency” behavior was recorded 24 times in the AI group versus 5 times in the control group. Similarly, “feedback confusion” (e.g., misinterpreting AI-generated code or mismanaging error output) occurred 20 times in the AI group and only 2 times in the control group. The “strategic rigidity” count was 12 for the AI group versus 2 for the control group. These behavioral discrepancies suggest that current AI systems may exacerbate cognitive conflict and task execution challenges within the creative environment.

A chi-square test indicated statistically significant group differences across seven of the eight behavioral dimensions, excluding submission timing (χ^2^_(6)_ = 42.83, *p* < 0.001). These findings suggest that AI intervention altered not only user–tool interaction, but also behavioral strategies and emotional regulation mechanisms.

Specifically, the AI group demonstrated a higher frequency of programming-related inquiries (23 vs. 8). Participant B-01 repeatedly experienced a loop of “AI output misinterpretation → seeking instructor support → regenerating output”, ultimately leading to intense frustration, accompanied by verbal and physical emotional expressions such as sighing and head-holding.

Regarding emotional regulation, the AI group demonstrated 17 instances of emotional reactivity (e.g., muttering, giving up, and expressing resistance), compared to only 5 in the control group. Participant B-06 explicitly stated that “the AI is useless” and exhibited ongoing resistance during the latter phase of the task. Participant B-08 abandoned the task after repeated AI output errors and submitted their work the earliest among all participants, representing a typical case of emotional collapse and task withdrawal.

In terms of task strategy, the AI group frequently exhibited “persistent debugging without strategy change” (n = 12), characterized by repetitive AI output generation without parameter adjustment or alternative planning. Participant B-12 exemplified this behavior, continuously generating identical command strings despite repeated error messages, indicating strategic rigidity and inefficient persistence. In contrast, the control group exhibited more flexible strategies, relying on course materials and reference samples to engage in trial-and-error adjustments.

The control group also outperformed the AI group in “information seeking” (12 vs. 3). For example, A-03 proactively consulted OpenProcessing tutorials when initially unclear about the task, ultimately building a coherent creative logic. Participant A-13 progressively developed a functional code block structure during early task phases, demonstrating strong integrative learning. Meanwhile, the AI group exhibited more instances of “cognitive dissociation” (e.g., distraction or confusion), with 14 occurrences versus 6 in the control group, suggesting greater susceptibility to cognitive overload and goal disengagement under AI conditions. For instance, participant B-15 frequently paused, stared blankly, and rechecked their AI output during the task, but ultimately failed to produce a structured outcome.

In contrast, the control group showed significantly better focus maintenance (19 vs. 5). Participant A-21 maintained stable progress throughout the session with minimal emotional fluctuation. The AI group exhibited both early exits and prolonged delays, resulting in a significantly higher variance in submission times compared to the control group, indicating the disruptive effect of system uncertainty on time management. Although overall peer interaction was low in both groups, it was slightly higher in the control group (4 vs. 2), suggesting that the emphasis on human–machine interaction may have reduced collaborative engagement in the AI group.

In summary, both frequency data and case analysis indicate that AI-assisted creative tasks, in their current technical form, do not improve task strategy or emotional regulation at the behavioral level. Instead, they introduce maladaptive behavioral patterns such as heightened emotional reactivity, increased tool dependency, diminished cognitive flexibility, and impaired task control.

## 4. Discussion

This study examined the impact of AI code generators on university students’ psychological states, ANS function, and creative output quality in the context of a creative task. Using a multimodal approach integrating emotional assessments (PANAS), creative self-efficacy (CSES), subjective task load (NASA-TLX), heart rate variability (HRV), and behavioral observations, we tested the following four theory-driven hypotheses: (H1) AI-assisted creation increases psychological stress and impairs autonomic nervous system regulation; (H2) AI usage increases negative affect and reduces positive affect; (H3) AI use results in greater perceived workload and emotional effort; and (H4) AI usage lowers the quality of creative output.

First, HRV indicators revealed phase-dependent physiological differences. At the task midpoint (T2), the AI group had a significantly higher LF/HF ratio (*p* = 0.050, η^2^ = 0.099), along with reductions in RMSSD and HF, indicating sympathetic dominance and impaired ANS regulation, thus supporting H1. As an index of brain–heart interaction, HRV reflected simultaneous rises in emotional load and physiological stress in the AI environment in this study, providing a physiological dimension to the cognitive neuroscience perspective. Second, the AI group showed a statistically significant decrease in positive mood scores and a statistically significant increase in negative mood scores on the post-task PANAS measure (∆PA: -3.7 ± 2.6, ∆NA: +3.1 ± 2.2, *p* < 0.01), validating H2, which states that AI use induces higher levels of negative mood. This emotional disturbance may stem from the AI system’s opacity, frequent failures, and unpredictable outputs, which undermine users’ sense of control and violate their expectations. Behavioral observations further substantiated these findings by revealing frequent indicators of emotional agitation in the AI group, including sighing, self-directed speech, verbalized frustration, and premature task disengagement, which suggests a depletion of emotional regulatory capacity under AI-assisted conditions. In line with H3, the AI-assisted participants reported significantly higher levels of mental demand, effort, and frustration on the NASA-TLX. These results indicate that, despite their automation potential, AI tools may impose hidden cognitive and emotional costs. The elevated perceived workload suggests increased prefrontal engagement to manage uncertainty, error correction, and interpretability gaps inherent in AI-generated content. Frustration may further reflect a mismatch between user expectations and system feedback, consistent with emotional–cognitive conflict models. Together, these findings highlight the substantial subjective toll of navigating opaque AI systems during complex tasks. In addition, the final creation results showed that the AI group’s work scores were significantly lower than those of the control group in several dimensions, such as creative integrity, expressiveness, and technical execution (difference in scoring means >1.2, *p* < 0.05), which verified the H4 hypothesis that the AI system, in its current form, may instead limit creators’ proactive thinking and innovative expression. Although AI tools enhance surface-level generation speed, their cognitive burden and emotional side effects may hinder creators’ active engagement and deep processing. The dissociation between subjective effort and creative quality is notable. Although AI tools may increase perceived task difficulty and emotional load, they do not necessarily correlate directly with expert-rated output, suggesting a complex relationship between user experience and product quality, particularly in novice users with limited domain knowledge.

These findings align with previous theoretical frameworks. According to Inzlicht et al.’s Emotional–Cognitive Conflict Model [34], technological opacity and feedback unpredictability may activate the prefrontal–amygdala circuit, impairing emotional regulation and decision making. The high emotional volatility and cognitive strain observed in the AI group reflect this conflict. Moreover, our HRV findings are consistent with Thayer et al.’s Central Autonomic Network (CAN) model [4], which posits that HRV reflects integrated regulation via the prefrontal cortex, anterior cingulate, and amygdala. A reduced HRV among AI users may, thus, indicate disrupted top-down modulation of stress responses. Recent studies have also explored the affective implications of AI interactions in various task contexts. For example, it has been reported that AI-generated feedback on educational platforms heightened emotional stress and reduced intrinsic motivation among students [35]. Reliance on opaque AI decision systems has been found to be associated with increased anxiety and cognitive uncertainty [36]. Similarly, AI-assisted creativity tools, while helpful in procedural aspects, have been found to reduce confidence and increase emotional disengagement [37]. In addition, the limited explainability of AI outputs has been shown to exacerbate user stress and impair affect regulation [38]. These findings converge with our results and suggest that the emotional dysregulation and elevated physiological arousal observed in the AI group may be driven not only by task complexity, but also by the affective opacity of AI systems, thus reinforcing the neuropsychological burden of AI-supported cognitive environments. However, it is important to note that these neural and affective responses were observed in participants at the earliest stage of domain learning. The extent to which such mechanisms generalize to expert users remains an open question.

Furthermore, the creative performance findings align with the concept of “adaptive tension between AI and creativity” proposed by Frith [39] and Sternberg with Kaufman [40]. These scholars contend that although AI can alleviate low-level cognitive burdens, it may also erode individuals’ creative autonomy and agency, potentially diminishing final output quality in certain contexts. In our study, this mechanism was reflected in both evaluation results and behavioral data, including higher task dependency, information avoidance, and feedback confusion in the AI group. Notably, a significantly higher proportion of early submissions and task withdrawals was observed among AI participants, suggesting a propensity toward cognitive disengagement under technological strain. These findings collectively underscore that behavioral outcomes must be interpreted within a broader psychoneurocognitive framework, and specifically within the context of novice learners undergoing early-stage skill acquisition.

A key consideration in interpreting these findings is the novice status of our participants, who had no formal programming experience and were engaging with AI-assisted creation for the first time. Their cognitive and emotional responses must, therefore, be understood within a foundational learning context, where domain-specific schemas [15] and self-efficacy are still under development [41]. Prior studies have shown that the psychological demands and feedback-related uncertainty of AI systems may be amplified in novice users, potentially leading to overreliance, confusion, and reduced intrinsic motivation [2]. As such, the emotional strain, physiological arousal, and reduced creative performance observed in this study should not be overgeneralized to experienced programmers or professional creators, who possess more stable domain knowledge and can exert greater control over AI output [20]. These results instead reflect a context-specific interaction between novice learning status and opaque AI feedback [42], underscoring the need for differentiated design and deployment strategies based on user expertise.

Despite the study’s systematic evaluation of the effects of AI code generation on emotion, self-efficacy, HRV, physiological stress, and behavioral performance, several limitations remain. First, although HRV is widely used as a proxy for autonomic nervous system activity, its accuracy and interpretability are partially constrained by the absence of complementary physiological parameters such as respiratory rate, electrodermal activity, and blood pressure. Prior research has shown that a reduced respiratory rate can lead to elevated LF power, potentially confounding the interpretation of LF/HF ratios [43]. Hence, the observed sympathetic activation in this study may have been partially influenced by variations in respiratory rhythms. Future research should incorporate respiratory and electrodermal metrics to offer a more comprehensive understanding of HRV dynamics. Second, the sample was composed exclusively of university students in design disciplines, limiting generalizability. The AI tool used was a single general-purpose Chinese-language model, without cross-platform comparisons. Third, although the sample size was determined based on a priori power analysis, it may still be underpowered to detect small or between-group effects, particularly in secondary outcome variables. This limitation should be taken into account when interpreting the robustness and generalizability of the findings. Furthermore, the participant population was highly gender-skewed, with 51 females and only 7 males. While the primary aim of this study was not to examine sex-based differences, this imbalance may limit the generalizability of the findings and precludes any meaningful analysis of sex-dependent responses. Future studies should aim for a more balanced sample distribution to assess potential gender effects. Individuals from other disciplines or using different AI systems may exhibit diverse emotional and behavioral responses to AI-assisted creation. Expanding sample diversity and adopting multi-platform experimental designs in future studies would improve both external and ecological validity. In addition, although participants’ emotional state was assessed using the PANAS at baseline, this measure primarily captures transient affective states rather than stable affective traits. Given that university students are known to exhibit elevated rates of subclinical anxiety and depression symptoms during academic activities [44,45], failure to screen for these conditions may confound the interpretation of affective and physiological outcomes. Future studies should consider incorporating standardized clinical screening tools (e.g., GAD-7 and PHQ-9) to distinguish between state-based emotional changes and trait-level affective predispositions. Lastly, behavioral observation relied predominantly on manual annotations by researchers. While a standardized coding protocol was employed, subjectivity and omission risks remain. Future studies should integrate objective behavioral tracking technologies, such as screen recording, keystroke/mouse logging, and eye-tracking systems, to construct high-resolution behavioral datasets and enable the fine-grained modeling of decision paths and psychophysiological mechanisms during task execution.

## 5. Conclusions

This study systematically examined the psychological, autonomic, and creative effects of AI code generators in the context of university students engaged in creative tasks. By integrating subjective emotional assessments (PANAS), creative self-efficacy measures (CSES), perceived task load evaluations (NASA-TLX), heart rate variability (HRV) data, and behavioral observations, the study identified consistent patterns indicating increased emotional strain, reduced autonomic flexibility, and diminished creative output under AI-assisted conditions.

Specifically, the participants in the AI-assisted group exhibited significantly lower positive affect and higher negative affect following task completion, indicating impaired emotional regulation. Physiologically, the AI group demonstrated reduced RMSSD and increased LF/HF ratios during the mid-task phase, reflecting heightened sympathetic nervous system activity and reduced parasympathetic modulation. This pattern indicates a decline in autonomic flexibility and impaired stress recovery capacity. Moreover, creative outputs from the AI-assisted group received significantly lower scores in dimensions such as originality, completeness, and aesthetic expression, suggesting that excessive reliance on AI may impair individuals’ capacity for active information processing and creative articulation in complex creative contexts. 

Although these results offer meaningful insights into the cognitive and affective implications of AI-assisted creation, they must be interpreted within the context of novice learners. All participants in this study were undergraduate design students with no formal training in programming, and their interaction with AI tools occurred during the initial phase of domain-specific skill acquisition. At this early stage, learners tend to lack established problem-solving schemas and self-efficacy, which may amplify the emotional and cognitive challenges posed by opaque and unpredictable AI feedback. Therefore, the observed effects should not be generalized to experienced users or professional developers. Rather, these findings highlight the importance of aligning AI system design with users’ developmental readiness and cognitive maturity.

Overall, the study contributes to the growing body of empirical research on human–AI interaction, especially within the domain of social cognitive and affective neuroscience. It proposes a multidimensional framework that integrates subjective experiences, behavioral indicators, and autonomic nervous system responses to elucidate the cognitive–affective–physiological coupling mechanisms at play under AI-assisted conditions. Future research should extend this work to more diverse populations and task environments, incorporating multimodal neurophysiological data to further explore the adaptive and maladaptive dynamics of cognitive and emotional regulation in AI-mediated contexts.

## Figures and Tables

**Figure 1 brainsci-15-00565-f001:**
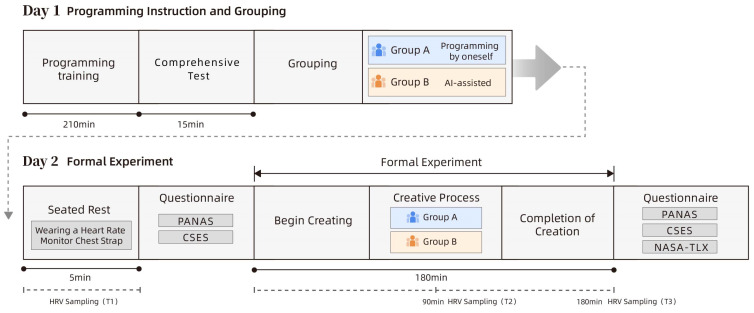
Experimental procedure. Solid arrows represent the temporal flow of tasks; the gray dashed arrow indicates the transition between Day 1 and Day 2. HRV = Heart Rate Variability. PANAS = Positive and Negative Affect Schedule. CSES = Creative Self-Efficacy Scale. NASA-TLX = NASA Task Load Index. Time is measured in minutes (min).

**Figure 2 brainsci-15-00565-f002:**
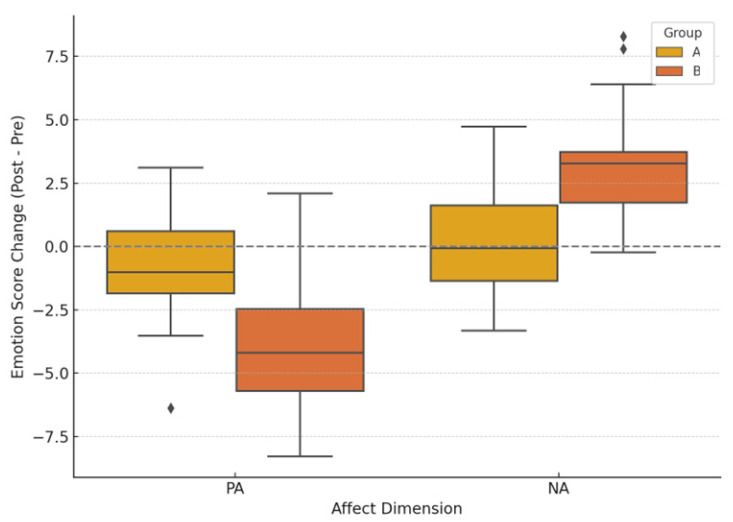
Boxplot of changes in positive (PA) and negative (NA) affect for the control group (A) and the AI-assisted group (B). Values represent post–pre difference scores. The gray dashed line indicates no change from baseline. The horizontal line inside each box indicates the median; boxes represent the interquartile range (IQR), whiskers extend to 1.5 × IQR. Diamonds represent outlier values beyond this range.

**Figure 3 brainsci-15-00565-f003:**
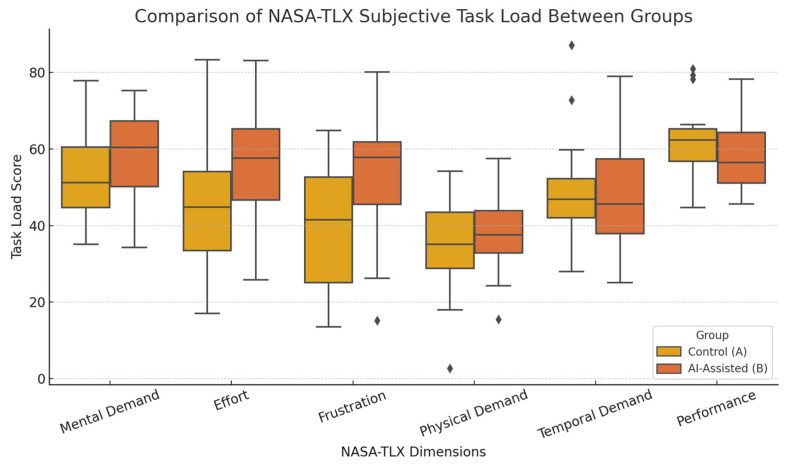
Boxplot comparison of subjective task load ratings across six NASA-TLX dimensions between the control group (A) and the AI-assisted group (B). Each box shows the interquartile range (IQR), the line inside the box represents the median, and whiskers extend to 1.5 times the IQR. Diamonds represent outlier values beyond this range. The color coding of groups has been revised to ensure consistency with the legend.

**Figure 4 brainsci-15-00565-f004:**
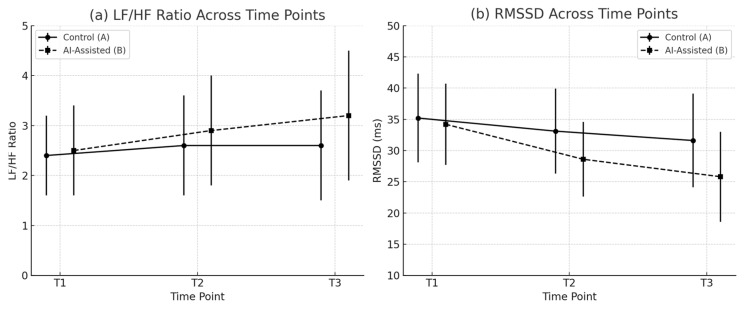
Group-wise HRV changes across task phases. (**a**) Line plot of LF/HF ratio across task phases (T1–T3), with separate trends for the control group and the AI-assisted group. (**b**) Line plot of RMSSD values across the three phases, showing a sharper decline in the AI-assisted group. Error bars denote ±1 SD.

**Figure 5 brainsci-15-00565-f005:**
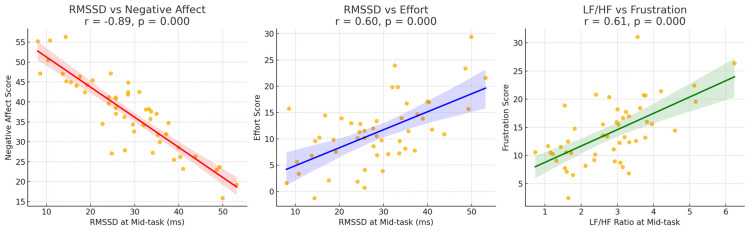
Scatterplot and correlation analysis of HRV physiological indicators (RMSSD vs. LF/HF) with subjective variables (negative mood, effort, and frustration). Regression lines with 95% CI are shown.

**Table 1 brainsci-15-00565-t001:** Group differences in creative scores and submission timing (mean ± SD, significance tests).

	Group A		Group B		*t*	*p*
	Mean	SD	Mean	SD		
Creativity Score	27.40	5.3	23.9	5.9	2.67	0.01
Total Score	76.54	17.61	72.19	18.51	2.10	0.04
Submission Time	175.1	6.2	162.3	22.7	-	0.007

**Table 2 brainsci-15-00565-t002:** Between-group comparison of pre-task and change scores on PANAS dimensions (mean ± SD).

Affect Dimension	Metric	Group A (*M* ± *SD*)	Group B (Δ Score)	*t* _(df)_	*p*-Value	Cohen’s d
Positive Affect (PA)	Pre-task Score	3.12 ± 0.90	2.97 ± 0.49	0.79(43)	0.435	-
	Δ Score (Post–Pre)	−0.7 ± 2.1	−3.7 ± 2.6	−2.14(58)	0.037	0.57
Negative Affect (NA)	Pre-task Score	2.77 ± 0.96	2.75 ± 0.66	0.09(50)	0.927	-
	Δ Score (Post–Pre)	+0.3 ± 1.9	+3.1 ± 2.2	2.86(58)	0.006	0.75

**Table 3 brainsci-15-00565-t003:** RM-ANCOVA results for LF/HF and RMSSD.

HRV Index	Effect	*F* _(df)_	*p*-Value	η^2^
LF/HF	Group	F(1,58) = 4.069	0.050 *	0.099
	Time	F(2,116) = 1.283	0.281	0.022
	Group × Time	F(2,116) = 2.113	0.125	0.036
RMSSD	Group	F(1,58) = 2.519	0.121	0.064
	Time	F (2,116) = 7.803	0.001 **	0.121
	Group × Time	F(2,116) = 3.285	0.043 *	0.082

*p* < 0.05 marked with *, *p* < 0.01 marked with **.

**Table 4 brainsci-15-00565-t004:** Item-level comparison of creative self-efficacy (CSES) scores before and after the task in the AI-assisted and control groups.

Item	Group	Pre-Test	Post-Test	ΔScore	*p*	*d*
1. Confidence in using creativity to solve problems.	AI Group	4.10 ± 0.64	3.67 ± 0.72	−0.43	0.018 *	0.65
	Control Group	3.90 ± 0.58	3.88 ± 0.60	−0.02	0.785	0.04
2. Good at coming up with new ideas.	AI Group	3.90 ± 0.60	3.53 ± 0.65	−0.37	0.023 *	0.61
	Control Group	3.80 ± 0.63	3.75 ± 0.59	−0.05	0.644	0.08
3. Skilled in developing ideas from others.	AI Group	4.00 ± 0.59	3.60 ± 0.62	−0.40	0.012 *	0.66
	Control Group	3.95 ± 0.55	3.88 ± 0.58	−0.07	0.552	0.12
4. Good at finding new ways to solve problems.	AI Group	4.07 ± 0.66	3.63 ± 0.68	−0.44	0.015 *	0.63
	Control Group	3.75 ± 0.61	3.78 ± 0.57	+0.03	0.772	0.06

*p* < 0.05 marked with *.

**Table 5 brainsci-15-00565-t005:** Comparative behavioral observations between AI-assisted and control groups.

No.	Behavior Category	Control Group	AI Group	Remarks
1	Tool Dependency	5	24	Significantly higher in AI group
2	Feedback Confusion	2	20	AI participants showed more frequent confusion
3	Strategic Rigidity	2	12	Persistent debugging without adjustment in AI group
4	Emotional Reactivity	5	17	Stronger emotional responses in AI group
5	Task Withdrawal	1	6	Early termination more common in AI group
6	Information Seeking	12	3	Control group showed higher proactive learning
7	Sustained Focus	19	5	Control group showed more stable task pacing
8	Peer Interaction	4	2	More collaboration in control

## Data Availability

The data presented in this study are available on request from the corresponding author. The data are not publicly available since they constitute an excerpt of research in progress.

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
