# Peer review of "The Neurophysiological Paradox of AI-Induced Frustration: A Multimodal Study of Heart Rate Variability, Affective Responses, and Creative Output"

_brainsci, 2025, doi:10.3390/brainsci15060565_

Round 1
Reviewer 1 Report
Comments and Suggestions for Authors
The topic of the study is certainly relevant, as artificial intelligence technologies have been developing very rapidly in recent time. The impact of AI on psychological state is not studied at all. The introduction describes the research problem clearly. In the methods, the researchers excluded intervening factors by anonymizing the results of the study. The results and their discussion are well written and detailed. The conclusions are consistent with the results and show perspectives and direction for future research. Overall this is a very interesting and relevant work.
The lack of information about the emotional state of the subjects prior to inclusion in the study is worth mentioning as shortcomings of this study. It is described that a sufficient percentage of healthy individuals, especially university students, who are stressed during their studies have anxiety and depressive symptoms. The authors should mention this in the limitations of the study. Also, few studies by other authors are described in the discussion. I recommend the authors to add to their discussion articles by other authors about the effect of AI on emotional state, symptoms of anxiety and depression, etc.
Author Response
Although I have submitted the “Author's Notes to Reviewer” through the manuscript system as required, I have also prepared a more detailed and clearly formatted version of my responses to the reviewers’ comments for your convenience.
Please kindly refer to the attached Author_response_to_review_comments.pdf, which outlines all revisions and point-by-point responses to the reviewers' suggestions. I hope this document will facilitate the review process and make my changes easier to track.
Comment 1:
The lack of information about the emotional state of the subjects prior to inclusion in the study is worth mentioning as a shortcoming of this study. It is described that a sufficient percentage of healthy individuals, especially university students, who are stressed during their studies have anxiety and depressive symptoms. The authors should mention this in the limitations of the study.
Response 1:
We appreciate the reviewer’s insightful observation regarding the baseline emotional state of participants. In our study, we assessed participants’ emotional states at baseline using the Positive and Negative Affect Schedule (PANAS), which was administered prior to the experimental task. However, we acknowledge that PANAS primarily captures transient, state-level affective responses and does not serve as a clinical diagnostic tool for chronic affective conditions such as anxiety or depression.
Recognizing that university students often exhibit subclinical levels of anxiety and depression during their academic careers (Eisenberg et al., 2007), we have addressed this limitation in the revised manuscript. Specifically, we added the following statement to the “Limitations” subsection of the Discussion:
[In addition, although participants’ emotional state was assessed using the PANAS at baseline, this measure primarily captures transient affective states rather than stable affective traits. Given that university students are known to exhibit elevated rates of subclinical anxiety and depression symptoms during academic activities (Eisenberg, D., et al; ), failure to screen for these conditions may confound the interpretation of affective and physiological outcomes. Future studies should consider incorporating standardized clinical screening tools (e.g., GAD-7, PHQ-9) to distinguish between state-based emotional changes and trait-level affective predispositions.]
Additionally, in Section 2.3 "Measurement," we clarified the usage of PANAS by adding:
[It should be noted that the PANAS measures state-based emotional responses and is not designed to diagnose clinical emotional disorders such as anxiety or depression.]
We have now included a reference to prior epidemiological data indicating the prevalence of anxiety and depressive symptoms in student populations.
[1. Eisenberg D, Gollust SE, Golberstein E, Hefner JL. Prevalence and correlates of depression, anxiety, and suicidality among university students. Am J Orthopsychiatry. 2007 Oct;77(4):534-42. doi: 10.1037/0002-9432.77.4.534.
2. Conteh I, Yan J, Dovi KS, Bajinka O, Massey IY, Turay B. Prevalence and associated influential factors of mental health problems among Chinese college students during different stages of COVID-19 pandemic: A systematic review. Psychiatry Res Commun. 2022 Dec;2(4):100082. doi: 10.1016/j.psycom.2022.100082.]
Comment 2:Also, few studies by other authors are described in the discussion. I recommend the authors to add to their discussion articles by other authors about the effect of AI on emotional state, symptoms of anxiety and depression, etc.
Response 2:Thank you for this helpful suggestion. We have significantly expanded the Discussion section to include recent literature addressing the emotional and psychological effects of AI systems. This includes empirical studies demonstrating the link between AI-induced cognitive uncertainty, emotional disengagement, and anxiety. We believe this addition reinforces the theoretical foundation of our findings and broadens the context of our results.
[In addition, although participants’ emotional state was assessed using the PANAS at baseline, this measure primarily captures transient affective states rather than stable affective traits. Given that university students are known to exhibit elevated rates of subclinical anxiety and depression symptoms during academic activities, failure to screen for these conditions may confound the interpretation of affective and physiological outcomes. Future studies should consider incorporating standardized clinical screening tools (e.g., GAD-7, PHQ-9) to distinguish between state-based emotional changes and trait-level affective predispositions.]
New references discussed include:
[1. Delello, J.A.; Sung, W.; Mokhtari, K.; Hebert, J.; Bronson, A.; De Giuseppe, T. AI in the Classroom: Insights from Educators on Usage, Challenges, and Mental Health. Educ. Sci. 2025, 15, 113. https://doi.org/10.3390/educsci15020113
2. Yuan, H. Artificial intelligence in language learning: biometric feedback and adaptive reading for improved comprehension and reduced anxiety. Humanit Soc Sci Commun 12, 556 (2025). https://doi.org/10.1057/s41599-025-04878-w
3. Wu, S., Liu, Y., Ruan, M. et al. Human-generative AI collaboration enhances task performance but undermines human’s intrinsic motivation. Sci Rep 15, 15105 (2025). https://doi.org/10.1038/s41598-025-98385-2
4. Smith K. Khare, Victoria Blanes-Vidal, Esmaeil S. Nadimi, and U. Rajendra Acharya. 2024. Emotion recognition and artificial intelligence: A systematic review (2014–2023) and research recommendations. Inf. Fusion 102, C (Feb 2024). https://doi.org/10.1016/j.inffus.2023.102019 ]

Reviewer 2 Report
Comments and Suggestions for Authors
This is an excellent paper and there is very little for a reviewer to report. The incorporation of physiological measures is particularly noteworthy and commendable.
The paper notes that the participant population had 7 male and 51 females. It’s difficult to speculate if this is significant. With such a small number of males an investigation of sex dependence is not possible. The imbalance in the participant population should be noted in the summary of the study’s limitations.
When reduced to fit the journal format Figure 4 is very difficult to read, even with a magnifying glass. Perhaps different colors could be used for the AI-group and the control group.
A word is misspelled in line 464.
Author Response
Although I have submitted the “Author's Notes to Reviewer” through the manuscript system as required, I have also prepared a more detailed and clearly formatted version of my responses to the reviewers’ comments for your convenience.
Please kindly refer to the attached Author_response_to_review_comments.pdf, which outlines all revisions and point-by-point responses to the reviewers' suggestions. I hope this document will facilitate the review process and make my changes easier to track.
Comment 1:The paper notes that the participant population had 7 males and 51 females. It’s difficult to speculate if this is significant. With such a small number of males an investigation of sex dependence is not possible. The imbalance in the participant population should be noted in the summary of the study’s limitations.
Response 1:Thank you for this important observation. We have now acknowledged the gender imbalance as a limitation of our study in the revised "Discussion" section. Specifically, we added: [Furthermore, the participant population was highly gender-skewed, with 51 females and only 7 males. While the primary aim of this study was not to examine sex-based differences, this imbalance may limit the generalizability of the findings and precludes any meaningful analysis of sex-dependent responses. Future studies should aim for a more balanced sample distribution to assess potential gender effects. ]
Comment 2: When reduced to fit the journal format Figure 4 is very difficult to read, even with a magnifying glass. Perhaps different colors could be used for the AI-group and the control group.
Response 2:
We sincerely thank the reviewer for the helpful suggestion regarding the readability of Figure 4. As advised, we have updated the figure to improve visual clarity:
(1) In the revised version, Control Group (A) is represented by a solid black line with circular markers, and AI-Assisted Group (B) is represented by a dashed black line with square markers.
(2) Standard deviation error bars are now clearly indicated with horizontal caps at each time point (T1, T2, T3).
(3) To enhance visual differentiation and reduce overlap, we applied a slight horizontal offset between the two groups’ data points at each time point.
(4) The figure retains a monochromatic black-and-white scheme, which aligns with academic publishing standards and improves print clarity.
Comment 3:A word is misspelled in line 464.
Response 3: Thank you for your careful reading. We have thoroughly reviewed the sentence around line 464 but were unable to identify a misspelled word in the current version of the manuscript. It is possible that the error may have been automatically corrected during the document editing process. If the misspelling remains in your version or refers to a specific word we may have overlooked, we would be grateful if you could kindly indicate the exact word so we can revise it immediately.

Reviewer 3 Report
Comments and Suggestions for Authors
- The hypothesis should be further elaborated to better match the measures and findings. There is no clear mapping on how the 3 instruments, PANAS, NASA-TLX, and Creative Self-Efficacy Scale (CSES) support each working hypothesis.
- Furthermore, in line 262 authors state: "These results further support Hypothesis H3, indicating that while AI tools may enhance technical capabilities, they also increase users’ subjective workload and emotional cost during creative processes." but H3 states "(H3) AI assistance will reduce the overall quality of creative output." So, H3 should be more closely related to the measured effects.
- Are there any significant differences among groups on each PANAS construct? Such pre-measures should be provided on a table. Also, the comparisons among groups on pre-measures to assess if the differences among groups on pre-post measures were affected by having a different baseline
- The same idea for the NASA-TLX constructs' analyses; pre-measures comparisons among groups are not provided, but are relevant to strengthen the elevated differences among groups or otherwise to smooth them.
- also for the Creative Self-Efficacy Analysis, especially item 4
- main limitations to incorporate are 1) the sample size and 2) its predominance of females over males.
Author Response
Although I have submitted the “Author's Notes to Reviewer” through the manuscript system as required, I have also prepared a more detailed and clearly formatted version of my responses to the reviewers’ comments for your convenience.
Please kindly refer to the attached Author_response_to_review_comments.pdf, which outlines all revisions and point-by-point responses to the reviewers' suggestions. I hope this document will facilitate the review process and make my changes easier to track.
Comment 1: The hypothesis should be further elaborated to better match the measures and findings. There is no clear mapping on how the 3 instruments, PANAS, NASA-TLX, and Creative Self-Efficacy Scale (CSES) support each working hypothesis.
Response 1:
We thank the reviewer for this insightful observation. In response, we have revised the structure of our hypotheses to improve conceptual clarity and better align with the distinct psychological and creative constructs assessed in the study. The updated version presents four theory-driven hypotheses, each corresponding to a major outcome domain: autonomic stress, emotional affect, perceived workload, and creative output.
Importantly, while we revised the framing of the hypotheses for clarity, we did not introduce any post hoc rationalization or exploratory reinterpretation of the data. The measurements, instruments, and analyses were all pre-specified prior to hypothesis testing. This update preserves the integrity of the hypothesis-testing framework and avoids the appearance of HARKing (Hypothesizing After the Results are Known), in line with good scientific practice. We included these clarifications in the introduction. Specifically, they are placed at the end of the section.
[In line with these objectives, we propose the following hypotheses:
H1: AI-assisted creation increases psychological stress and impairs autonomic nervous system regulation;
H2: AI assistance leads to higher levels of negative affect and lower levels of positive affect during the creative process;
H3: Participants who use AI report greater perceived cognitive workload and frustration compared to those in the control group;
H4: The creative output quality is lower among AI-assisted participants than among those who complete the task independently.]
We also incorporated the same clarifications in the measurement section, positioning them at the start of Section 2.3. [A multimodal measurement framework was employed to test the four hypotheses. Hypothesis 1 (H1), related to autonomic stress, was assessed using heart rate variability (HRV) indices RMSSD and LF/HF ratio collected at three task phases. Hypothesis 2 (H2), concerning affective changes, was evaluated using the Positive and Negative Affect Schedule (PANAS), administered before and after the task. Hypothesis 3 (H3) was tested using the NASA Task Load Index (NASA-TLX), which measures perceived mental demand, effort, and frustration. Hypothesis 4 (H4) was tested through expert blind evaluations of participants’ creative outputs, based on standardized rubrics of creativity, technical execution, and thematic relevance.]
Comment 2:Furthermore, in line 262 authors state: 'These results further support Hypothesis H3, indicating that while AI tools may enhance technical capabilities, they also increase users’ subjective workload and emotional cost during creative processes.' but H3 states '(H3) AI assistance will reduce the overall quality of creative output.' So, H3 should be more closely related to the measured effects.
Response 2: We thank the reviewer for this valuable observation. In response, we revised the structure of our hypotheses to clearly separate the effects of AI assistance on creative output and on subjective workload. Specifically, the original Hypothesis 3 has now been split into two distinct hypotheses: H3, which addresses increased perceived cognitive and emotional load, and H4, which concerns reduced quality of creative output. This structural revision was made to ensure that each hypothesis is closely aligned with the specific measures used in the study and that their interpretation remains theoretically consistent.
[we tested four theory-driven hypotheses: (H1) AI-assisted creation increases psychological stress and impairs autonomic nervous system regulation; (H2) AI usage increases negative affect and reduces positive affect; (H3) AI use results in greater perceived workload and emotional effort; and (H4) AI usage lowers the quality of creative output.
......
Behavioral observations further substantiated these findings by revealing frequent indicators of emotional agitation in the AI group, including sighing, self-directed speech, verbalized frustration, and premature task disengagement, which suggests a depletion of emotional regulatory capacity under AI-assisted conditions. In line with H3, AI-assisted participants reported significantly higher levels of mental demand, effort, and frustration on the NASA-TLX. These results indicate that, despite their automation potential, AI tools may impose hidden cognitive and emotional costs. The elevated perceived workload suggests increased prefrontal engagement to manage uncertainty, error correction, and interpretability gaps inherent in AI-generated content. Frustration may further reflect a mismatch between user expectations and system feedback, consistent with emotional-cognitive conflict models. Together, these findings highlight the substantial subjective toll of navigating opaque AI systems during complex tasks. ]
Based on this revised hypothesis framework, we have substantially rewritten the Discussion section to ensure each hypothesis is interpreted in relation to its corresponding results. The previously cited sentence at line 262 has been removed, as it conflated subjective workload with creative performance.
To clarify this distinction, we have also added a new paragraph in the Discussion highlighting the conceptual dissociation between user experience and creative outcomes. The following sentence now appears in the revised text:
[The dissociation between subjective effort and creative quality is notable. Although AI tools may increase perceived task difficulty and emotional load, they do not necessarily correlate directly with expert-rated output, suggesting a complex relationship between user experience and product quality.]
Comment 3: Are there any significant differences among groups on each PANAS construct? Such pre-measures should be provided on a table. Also, the comparisons among groups on pre-measures to assess if the differences among groups on pre-post measures were affected by having a different baseline.
Response 3:
Thank you for highlighting this important point. In response, we conducted additional independent-sample t-tests to compare the AI-assisted group and the control group on pre-task PANAS scores (Positive Affect and Negative Affect), in order to examine potential baseline differences.
The results indicated no significant group differences in either PA (Control: 3.12 ± 0.90; AI: 2.97 ± 0.49; t(43) = 0.79, p = 0.435) or NA (Control: 2.77 ± 0.96; AI: 2.75 ± 0.66; t(50) = 0.09, p = 0.927), confirming that the two groups were affectively comparable at baseline.
To reflect this analysis, we made the following changes in the manuscript:
A new sentence was added at the beginning of Section 3.2 to report the baseline comparison and clarify that the observed post-task differences are unlikely to be confounded by pre-existing affective differences.
[To ensure baseline equivalence, independent-sample t-tests were first conducted on pre-task PANAS scores. Degrees of freedom were estimated using the Welch–Satterthwaite equation. The results showed no significant group differences in either PA or NA prior to the task, with PA scores of 3.12 ± 0.90 for the control group and 2.97 ± 0.49 for the AI group, t(43) = 0.79, p = 0.435; and NA scores of 2.77 ± 0.96 vs. 2.75 ± 0.66, t(50) = 0.09, p = 0.927, respectively. These findings confirm that the two groups were comparable in affective state at baseline (see Table 2).]
Also, Table 2 was revised to integrate both the pre-task comparison results and the post–pre emotional change scores into a single, comprehensive summary table.
Comment 4: The same idea for the NASA-TLX constructs' analyses; pre-measures comparisons among groups are not provided, but are relevant to strengthen the elevated differences among groups or otherwise to smooth them.
Respomse 4:
Thank you for raising this point. We fully agree with the reviewer that baseline comparability is critical for interpreting between-group differences. However, the NASA-TLX is a task-specific, post-task self-report instrument designed to assess perceived workload after task engagement. Therefore, pre-task comparisons are not applicable in this context, as participants had no basis to assess mental demand, effort, or frustration before completing the creative task.
That said, we acknowledge the importance of ruling out potential confounds due to pre-existing individual differences. To address this concern indirectly, we ensured that participants were randomly assigned to experimental conditions, and that both groups were comparable in demographic variables and PANAS pre-task affective states (see Table 2). These factors help reduce the risk that differences in NASA-TLX scores were driven by individual predispositions rather than by the task experience itself.
We have clarified this point in the manuscript by adding a brief methodological note in the Results section where NASA-TLX findings are introduced.
[As NASA-TLX is a post-task instrument designed to capture subjective workload following task completion, no pre-task scores are available. However, random assignment and comparable pre-task affective states across groups reduce the likelihood that observed differences stemmed from individual predispositions.]
Comment 5:also for the Creative Self-Efficacy Analysis, especially item 4.
Response 5:
Thank you for drawing attention to this important detail. We have carefully reviewed the item-level results of the Creative Self-Efficacy Scale (CSES), with particular focus on item 4 (“Good at finding new ways to solve problems”), which is especially relevant to the central theme of AI-supported creative autonomy.
In our results, item 4 showed a statistically significant decline in the AI group (pre-task M = 4.07 ± 0.66; post-task M = 3.63 ± 0.68; p = 0.015, d = 0.63), indicating that participants felt less confident in their ability to independently solve problems after interacting with the AI tool. This effect was not observed in the control group, which instead showed a small, non-significant increase. These findings suggest that AI assistance may impair users’ perceived creative agency, even when objective task completion is supported.
To reflect this observation, we have added a new sentence in Section 3.5 of the Results that highlights the theoretical relevance of item 4 and its implications for perceived creative autonomy. We have also retained the full item-level breakdown in Table 4, ensuring transparency and accessibility for readers interested in this construct.
[Among these, item 4 (“Good at finding new ways to solve problems”) is particularly noteworthy, as it directly reflects participants’ confidence in approaching problems creatively and independently. The AI group showed a significant decline on this item (p = 0.015, d = 0.63), which may indicate a perceived erosion of creative agency in the presence of AI assistance. By comparison, the control group exhibited a slight, non-significant increase on this same item (Δ = +0.03, p = 0.772, d = 0.06), suggesting possible motivational benefits of self-reliant engagement.]
Comment 6: main limitations to incorporate are 1) the sample size and 2) its predominance of females over males.
Response 6:
Thank you for this important comment. In response, we have addressed both aspects in the revised manuscript:
1. Sample size — We have added a description of our a priori power analysis in the Participants section. Specifically, we used G*Power 3.1 to estimate the required sample size to detect medium-to-large effects (d = 0.63) with 80\% power and α = 0.05, which indicated that approximately 23 participants per group would be needed for within-group comparisons.
[A priori power analysis was conducted using G*Power 3.1 to estimate the required sample size for detecting medium-to-large effects (d = 0.63) with 80\% power and a two-tailed significance level of 0.05. The analysis indicated that approximately 23 participants per group would be required under these parameters. Each group in this study initially included 29 participants, exceeding this reference value.]
We have also acknowledged in the Discussion section that while our sample size was planned based on this analysis, it may limit statistical power for detecting small effects, especially those arising from between-group comparisons, and this has been noted as a limitation of the study.
[Third, although the sample size was determined based on a priori power analysis, it may still be underpowered to detect small or between-group effects, particularly in secondary outcome variables. This limitation should be taken into account when interpreting the robustness and generalizability of the findings.]
2. Gender imbalance — The gender distribution of our sample (51 females, 7 males) was already discussed in the original version of the manuscript. In the revised version, we retained and slightly expanded this discussion to emphasize that the predominance of female participants limits the generalizability of the findings and precludes the investigation of sex-based differences.
[Furthermore, the participant population was highly gender-skewed, with 51 females and only 7 males. While the primary aim of this study was not to examine sex-based differences, this imbalance may limit the generalizability of the findings and precludes any meaningful analysis of sex-dependent responses. Future studies should aim for a more balanced sample distribution to assess potential gender effects. ]

Reviewer 4 Report
Comments and Suggestions for Authors
- The primary aim of this study is to investigate 73 the role of AI code generators in artistic creative tasks, specifically whether such tools 74 affect creators’ emotional experience, stress levels, and creative performance. While prior research is mentioned, the literature review does not clearly identify the gap this study addresses. The authors are encouraged to explicitly state what is missing in existing studies and how their work fills this void.
- The sample size (N=58) is relatively limited for an experimental design of this complexity. Given the use of multiple psychological and physiological measures, there may be insufficient statistical power to detect meaningful effects. It is recommended that the authors provide a power analysis or discuss this limitation.
- The strong gender imbalance (51 females, 7 males) may reduce the generalizability of the findings. The authors should consider discussing how this imbalance might
- The authors should determine the inclusion and exclusion criteria clearly.
- The manuscript should include a table that includes the Cronbach's alpha values, means, standard deviations, and minimum and maximum scores for all scales and dimensions respectively (PANAS, NASA-TLX, CSES).
- Please give detail which statistical nethods use to check normality assumptions and report reults for normality test
Author Response
Although I have submitted the “Author's Notes to Reviewer” through the manuscript system as required, I have also prepared a more detailed and clearly formatted version of my responses to the reviewers’ comments for your convenience.
Please kindly refer to the attached Author_response_to_review_comments.pdf, which outlines all revisions and point-by-point responses to the reviewers' suggestions. I hope this document will facilitate the review process and make my changes easier to track.
Comment 1: While prior research is mentioned, the literature review does not clearly identify the gap this study addresses. The authors are encouraged to explicitly state what is missing in existing studies and how their work fills this void.
Response 1: Thank you for this helpful suggestion. In response, we have revised the final part of the Introduction to more explicitly define the research gap. While earlier studies have examined either productivity or emotional aspects of AI-assisted work, few have investigated the combined emotional, physiological, and behavioral consequences of using AI tools in open-ended creative tasks. We now clearly state that existing research often relies solely on self-report data and lacks multimodal validation, particularly from a psychophysiological perspective. A new transition sentence has been added to underscore the need for such an integrated approach.
[Despite the growing interest in AI-assisted creativity, most existing studies have focused on either productivity outcomes or user perceptions in isolation, and often rely solely on self-report data without physiological validation. There remains a lack of research that systematically investigates the emotional, physiological, and behavioral consequences of AI use in open-ended creative tasks, especially from a psychophysiological perspective. Such an approach is necessary to gain a more comprehensive understanding of how AI systems shape users’ internal states and behavioral outcomes during creative engagement.
To address this gap...]
Comment 2: The sample size (N=58) is relatively limited for an experimental design of this complexity. Given the use of multiple psychological and physiological measures, there may be insufficient statistical power to detect meaningful effects. It is recommended that the authors provide a power analysis or discuss this limitation.
Response 2: Thank you for this important comment. In response, we have added a description of our a priori power analysis in the Participants section. Specifically, we used G*Power 3.1 to estimate the required sample size to detect medium-to-large effects (d = 0.63) with 80\% power and α = 0.05, which indicated that approximately 23 participants per group would be needed for within-group comparisons.
[A priori power analysis was conducted using G*Power 3.1 to estimate the required sample size for detecting medium-to-large effects (d = 0.63) with 80\% power and a two-tailed significance level of 0.05. The analysis indicated that approximately 23 participants per group would be required under these parameters. Each group in this study initially included 29 participants, exceeding this reference value.]
We have also acknowledged in the Discussion section that while our sample size was planned based on this analysis, it may limit statistical power for detecting small effects, especially those arising from between-group comparisons, and this has been noted as a limitation of the study.
[Third, although the sample size was determined based on a priori power analysis, it may still be underpowered to detect small or between-group effects, particularly in secondary outcome variables. This limitation should be taken into account when interpreting the robustness and generalizability of the findings.]
Comment 3: The strong gender imbalance (51 females, 7 males) may reduce the generalizability of the findings. The authors should consider discussing how this imbalance might.
Response 3: Thank you for highlighting this important point. In response, the gender distribution of our sample (51 females, 7 males) was already discussed in the original version of the manuscript. In the revised version, we retained and slightly expanded this discussion to emphasize that the predominance of female participants limits the generalizability of the findings and precludes the investigation of sex-based differences.
[Furthermore, the participant population was highly gender-skewed, with 51 females and only 7 males. While the primary aim of this study was not to examine sex-based differences, this imbalance may limit the generalizability of the findings and precludes any meaningful analysis of sex-dependent responses. Future studies should aim for a more balanced sample distribution to assess potential gender effects. ]
Comment 4: The manuscript should include a table that includes the Cronbach's alpha values, means, standard deviations, and minimum and maximum scores for all scales and dimensions respectively (PANAS, NASA-TLX, CSES).
Response 4: Thank you for your helpful suggestion. In response, we have created a comprehensive table that includes the internal consistency (Cronbach’s alpha), means, standard deviations, and minimum and maximum scores for each dimension of the PANAS, NASA-TLX, and CSES scales. To ensure clarity and avoid overloading the main text, this table has been placed in Appendix A and a description is provided in the manuscript. As the NASA-TLX and CSES instruments were administered using independent single-item dimensions, Cronbach’s α values were not calculated for these measures. Internal consistency was only computed for PANAS subscales, which involved multiple items per dimension.
[Descriptive statistics and reliability information (Cronbach’s α, means, standard deviations, and score ranges) for all measurement scales (PANAS, NASA-TLX, and CSES) are provided in Appendix A (Table A1).]
Comment 5:Please give detail which statistical nethods use to check normality assumptions and report reults for normality test.
Response 5: Thank you for pointing out the need to clarify the statistical methods used to check normality assumptions. We have now explicitly stated in the revised manuscript (Section 2.4, “Data Analysis”) that Shapiro–Wilk tests were used to assess the normality of all continuous variables. Based on the test results, we applied parametric or non-parametric methods accordingly. This addition improves the transparency and methodological rigor of the statistical analysis section.
[Before conducting inferential analyses, the normality of all continuous variables was assessed using the Shapiro–Wilk test. The results indicated that PANAS and CSES scores were normally distributed (p > 0.05), justifying the use of parametric tests. However, several NASA-TLX dimensions deviated significantly from normality (p < 0.05), and thus non-parametric Mann–Whitney U tests were employed for between-group comparisons on these subscales. ]

Round 2
Reviewer 4 Report
Comments and Suggestions for Authors
The paper has been properly revised, it can be accepted.
Author Response
Thank you for your positive evaluation. We appreciate your recognition that the paper has been properly revised. We are grateful for your guidance throughout the review process and are pleased that the manuscript now meets the publication standards of Brain Sciences. We look forward to the next steps.